# Differential Responders to a Mixed Meal Tolerance Test Associated with Type 2 Diabetes Risk Factors and Gut Microbiota—Data from the MEDGI-Carb Randomized Controlled Trial

**DOI:** 10.3390/nu15204369

**Published:** 2023-10-15

**Authors:** Viktor Skantze, Therese Hjorth, Mikael Wallman, Carl Brunius, Johan Dicksved, Erik A. Pelve, Anders Esberg, Marilena Vitale, Rosalba Giacco, Giuseppina Costabile, Robert E. Bergia, Mats Jirstrand, Wayne W. Campbell, Gabriele Riccardi, Rikard Landberg

**Affiliations:** 1Fraunhofer-Chalmers Research Centre for Industrial Mathematics, 412 88 Gothenburg, Swedenmats.jirstrand@fcc.chalmers.se (M.J.); 2Department of Life Sciences, Food and Nutrition Science, Chalmers University of Technology, 412 96 Gothenburg, Sweden; therese.hjorth@chalmers.se (T.H.); rikard.landberg@chalmers.se (R.L.); 3Department of Anatomy, Physiology and Biochemistry, Swedish University of Agricultural Sciences, 750 07 Uppsala, Sweden; 4Department of Animal Nutrition and Management, Swedish University of Agricultural Sciences, 750 07 Uppsala, Sweden; erik.pelve@slu.se; 5Department of Odontology, Umeå University, 901 87 Umeå, Sweden; 6Diabetes, Nutrition and Metabolism Unit, Department of Clinical Medicine and Surgery, Federico II University, 80138 Naples, Italy; marilena.vitale@unina.it (M.V.); rosalba.giacco@isa.cnr.it (R.G.); giuseppina.costabile@unina.it (G.C.); gabriele.riccardi@unina.it (G.R.); 7Institute of Food Sciences, National Research Council, 83100 Avellino, Italy; 8Department of Nutrition Science, Purdue University, West Lafayette, IN 47907, USAcampbeww@purdue.edu (W.W.C.)

**Keywords:** differential responders, clustering, personalized nutrition

## Abstract

The global prevalence of type 2 diabetes mellitus (T2DM) has surged in recent decades, and the identification of differential glycemic responders can aid tailored treatment for the prevention of prediabetes and T2DM. A mixed meal tolerance test (MMTT) based on regular foods offers the potential to uncover differential responders in dynamical postprandial events. We aimed to fit a simple mathematical model on dynamic postprandial glucose data from repeated MMTTs among participants with elevated T2DM risk to identify response clusters and investigate their association with T2DM risk factors and gut microbiota. Data were used from a 12-week multi-center dietary intervention trial involving high-risk T2DM adults, comparing high- versus low-glycemic index foods within a Mediterranean diet context (MEDGICarb). Model-based analysis of MMTTs from 155 participants (81 females and 74 males) revealed two distinct plasma glucose response clusters that were associated with baseline gut microbiota. Cluster A, inversely associated with HbA1c and waist circumference and directly with insulin sensitivity, exhibited a contrasting profile to cluster B. Findings imply that a standardized breakfast MMTT using regular foods could effectively distinguish non-diabetic individuals at varying risk levels for T2DM using a simple mechanistic model.

## 1. Introduction

The global prevalence of type 2 diabetes mellitus (T2DM) is expected to reach 783.2 million (12.2%) in 2045 [1]. T2DM is strongly associated with the risk of developing cardiovascular disease (CVD), which is the leading cause of mortality and morbidity globally [2,3]. Major risk factors for the development of T2DM include heredity, increased waist circumference, and elevated glycated hemoglobin (HbA1c). Furthermore, elevated glycemic variability and postprandial glucose and insulin responses may affect the risk of developing T2DM and CVD among non-diabetic individuals [4,5].

Recent studies have revealed large inter-individual variations in plasma glucose after corresponding meals and found that gut microbiota and food structure are important determinants of the differential response [6,7]. Hallmark studies have developed models to predict plasma glucose concentrations based on gut microbiota data, health information, and basic subject characteristics [6,8]. Personalized guidelines based on such predictions were shown to be more efficient than healthy dietary patterns, such as a common Mediterranean diet in lowering HbA1c levels among prediabetic patients [9], although large differences in total carbohydrate intake between the groups may in part have confounded the results. Thus, tailored guidelines based on the glucose variability in postprandial dynamic response may be an effective way to lower the risk of T2DM.

Clustering of dynamic features of the postprandial response could aid in identifying differences in glucose variability to the same dietary intake and provide a simple way of categorizing individuals according to diabetes risk. Such clusters may be targets for tailored diet and lifestyle interventions to prevent prediabetes or T2DM. Differential responders were identified after consuming an oral glucose tolerance test (OGTT) [10], but mixed meal tolerance tests (MMTT) based on regular foods are underexplored for this purpose when measuring only glucose [11]. Strong correlations between the response to OGTT and MMTT have been reported, showing that mixed meals with more complex composition that also affect lipid and protein metabolism are effective in reflecting glucose metabolism and perturbation thereof [11]. Furthermore, although gut microbiota was shown to be a key determinant of the inter-individual variation in postprandial glucose response, it has, to the best of our knowledge, not been associated with differential postprandial glucose response clusters [4,6,8].

Therefore, we aimed to investigate if a simple mechanistic model of glucose regulation could be applied to describe postprandial glucose concentrations after a standardized MMTT based on regular foods and whether clusters of differential responders could be identified from such a model. Furthermore, we investigated if differential response clusters are associated differently with risk factors of T2DM. We further investigated if baseline gut microbiota was associated with the response clusters and if clusters remained after dietary intervention with low or high glycemic index. The methodology was applied to data from non-diabetic men and women from Sweden, Italy, and the USA with overweight or obesity participating in interventions with high or low glycemic index (GI) Mediterranean diets [12,13].

## 2. Materials and Methods

### 2.1. Clinical Trial and Dietary Intervention

Data from the MEDGI-Carb trial were used in the present study because participants were at risk of developing T2DM, and the intervention tested the effect of a high vs. low GI diet within the context of a healthy eating pattern, i.e., a Mediterranean diet. By including individuals with elevated risk of T2D and using data from a dietary intervention with large contrasts in GI, we had the chance to evaluate the possibility of identifying glycemic response clusters across a wide range of likely postprandial glucose responses and to assess their stability during the intervention.

The MEDGI-Carb trial was an international multi-center randomized, controlled, parallel-group, 15-week dietary trial, including a 3-week baseline period followed by a 12-week controlled dietary intervention in adults at elevated risk of developing type 2 diabetes (give their age range, BMI range, OR provide a small table with their features (also, the other risk factors, up to you)). During the 12-week intervention period, participants consumed a Mediterranean-style, controlled, isocaloric, weight-maintenance diet. Furthermore, the participants were instructed to consume either a low-GI or high-GI diet with intervention-specific foods. All participants were instructed to consume the same amount of digestible carbohydrates (270 g/d) and dietary fiber (35 g/d). Modulation of daily energy intake was achieved by adjusting intakes of proteins and fat. Half of the daily carbohydrate intake was identical in the two groups, including vegetables and fruit. The other half consisted of carbohydrates with GI < 55 and >70 in the low and high GI groups, respectively. The intervention-specific carbohydrates were distributed throughout the day, with 26% at breakfast, 30% at lunch, and 44% at dinner. Markers of glucose homeostasis were obtained during standardized testing days by completion of an eight-hour MMTT, an OGTT, and 6 days of 24 h CGM at baseline and post-testing. Furthermore, blood samples were drawn to measure HbA1c, insulin, glucose, high-density lipoprotein, triglycerides, blood pressure, and anthropometrics. Insulin sensitivity indices such as the quantitative insulin sensitivity check index (QUICKI), Stumvoll, and Matsuda were calculated using data from the OGTTs [14]. The study was conducted at three centers: (i) Federico II University, Naples, Italy, (ii) Chalmers University of Technology, Gothenburg, Sweden, and (iii) Purdue University, West Lafayette, IN, USA. The study was initiated in January 2018, and the last participant finished the trial in December 2019. The trial was registered in the public trial registry clinicaltrials.gov as NCT03410719 prior to initiating participant recruitment. The study protocol was approved by the intuitional review boards at Purdue University and Federico II University and by the Swedish Ethical Review Authority. The study protocol with detailed descriptions of the trial, such as randomization, blood sampling, anthropometric measurements, and intervention diet, was previously published [12], as well as the results of the primary analysis [13].

### 2.2. Mixed Meal Tolerance Tests

Breakfast and lunch MMTT were performed at baseline, mid-testing (USA only), and post-intervention. Prior to all testing days, participants were instructed not to eat or drink anything (except a small amount of water) from 10:00 p.m., the evening before the visit. Fasting blood samples were collected at the time point (TP) −15 min and TP −5 following 15 min of rest. The test meal was consumed at TP 0 in two parts; the participants had 7.5 min to consume the first part of the meal and 7.5 min to consume the last part to control the pace of the meal consumption. The participants were allowed to drink 8 ounces of water (approx. 2.4 dL) during the meal. The test meals were strictly standardized across all three centers. All participants were served the same portion size, i.e., kilocalories, regardless of energy requirement for practical reasons. The food composition and nutrients of the standardized meals are provided in Table 1.

Blood samples were collected at TP 15 after the test meal and then at TP 30, TP 45, TP 60, TP 90, TP 120, TP 180, and TP 240. A standardized lunch meal was served at TP 240, again with 7.5 min to consume the first half of the meal and 7.5 min to consume the second part. The blood sampling continued by the same pattern as after the breakfast meal (Appendix A). For the present analysis, only glucose data from testing minutes TP −15 to TP 240 (i.e., the breakfast meal) were included to accommodate the fit of a simple kinetic model of the postprandial glucose response.

### 2.3. Oral Glucose Tolerance Test

Participants completed OGTT at baseline, mid-testing (USA only), and post-intervention. Fasting blood samples were collected at TP −15 after 15 min of rest and at TP −5. At TP 0, participants were instructed to consume a test beverage containing 75 g glucose dissolved in water within 5 min. No additional fluids were permitted during the test. Blood samples were collected at TP 60 and TP 120 (Appendix A).

### 2.4. Fecal Microbiota

During pre- and post-intervention study days, participants were asked to collect fecal samples using a stool sampling collection kit. Samples were taken using an EasySampler Stool Collector and a sample tube with a spoon lid. The sample was protected by being placed in yet another tube and stored immediately in the freezer (−20 °C). The samples were transported in a cooling box with an ice pack to the clinic within 72 h after the sample was collected. At the clinic, the sample is transferred to −80 °C within 24 h. The samples were analyzed with the 16S rRNA gene amplicon sequencing method, where DNA was extracted and purified from fecal samples using the QIAamp FAST DNA Stool mini kit from Qiagen, Venlo, The Netherlands. The DNA extraction followed the protocol from the manufacturer with one exception, where lysis of the bacterial cell walls used a mechanical lysis step (bead beating 2 × 1 min in a Precellys Evolution using 0.1 mm zirconium/silica beads). Once the DNA was extracted and purified from the sample, polymerase chain reaction (PCR) was used to amplify the V3-V4 region of the gene encoding 16S rRNA. This gene exists in all bacteria and is normally used for the taxonomic classification of bacteria since parts of the 16S gene vary in sequence composition between different bacteria. Sample-specific barcodes and Illumina adapters were then attached to the PCR amplicons to enable the pooling of samples. The library of PCR amplicons was then sequenced on the Illumina NovaSeq 6000 platform at Novogene, Singapore. The bioinformatics analysis to handle the generated sequence data used QIIME2 via the dada2 pipeline. The sequences were first demultiplexed, i.e., separated according to the sample-specific barcode in the specific sample. Then, quality control and filtration of the sequence data was performed to remove sequences with poor quality. Finally, a taxonomic classification of the sequences was performed. The gut microbiota analysis and subsequent data processing and analysis were described in detail by Iversen and Dicksved [15]. The gut microbiota was analyzed to investigate if it could explain inter-individual differences in glycemic postprandial response and provide some mechanistic insights into potential differential response profiles between subjects. The microbiota was aggregated to the genus level prior to downstream analysis.

Species that were known from the literature to associate with glucose regulation were selected to assess association with response clusters. The selected species were “*Bifidobacterium*”, “*Bacteroides*”, “*Faecalibacterium*”, “*Akkermansia*”, “*Roseburia*”, “*Fusobacterium*”, “*Blautia*”, “*Haemophilus*”, “*Ruminococcus*”, “*Clostridium*”, and “*Dorea*” [16,17,18,19,20,21,22].

### 2.5. Mechanistic Model of Glucose Regulation

A modified version of the minimal glucose model [23] proposed by Bolie [24] was used to describe the glucose response to the MMTT from breakfast at baseline and after 12 wk in order to identify interpretable parameters that could be used as the basis for grouping individuals according to plasma glucose–time profiles [25]. The model was initially developed for plasma glucose data following an OGTT and consists of two coupled differential equations describing the feedback loop of glucose and insulin blood concentrations in response to glucose intake [25]. In the present study, the MMTT consisted of a carbohydrate-rich meal, which was hypothesized to give a similar response to that of an OGTT when consumed at a fasted state, although it is acknowledged that the model is oversimplifying the complex glucose–insulin system in response to foods that also contain other nutrients and non-nutrients. The idea was not to provide a model that explains all biological processes related to the postprandial response to a mixed meal but rather to provide a simple model that fits the data and could be used to group responders into clusters that are differently related to T2D risk factors.

The dynamics of the model are described using compartments that represent mechanisms in the glucose–insulin system, and the exchange rates between compartments are described using rate constants. The model assumes that the ingested glucose is delayed by the digestive system and transferred to the bloodstream, where insulin acts to let the glucose be absorbed by the muscle tissue or the liver and converted to glycogen. Furthermore, the model assumes that the glucose can be discarded through the urine via the kidneys and that the pancreas produces insulin at a given rate in response to the current glucose concentration. Notably, the model assumes a linear relationship between insulin secretion and glucose, which is often not the case due to the effects of incretin hormones, for example, but the rationale is to obtain a more parsimonious model with easily grouped parameters. A particularly simple solution for the model glucose concentration can be formulated if the gastrointestinal absorption is assumed to rise very quickly and fall slowly and the ingested breakfast meal is modeled as a momentary impulse at the first measurement in time [25]. This solution takes the shape of a damped sinusoidal wave (Equation (1)), which is used widely in mechanics [26]. Thus, the parameters governing the glucose dynamics were reduced to a glucose baseline level ($G_{b}$), sinusoidal amplitude (A) involved in the resulting amplitude of the glucose concentration, sinusoidal frequency (ω) relating to the velocity of glucose oscillations, and damping coefficient (α) determining the rate of glucose decay.
(1)$$G{\left(t\right)}_{\;}=G_{b}+A\;\sin\left(\omega t\right)\;e^{-\alpha t}$$

Although the parameters of the reduced model have no one-to-one correspondence to specific mechanisms in the body, they convey the general quality of the glucose control.

The sinusoidal frequency (ω) relates to the rates of removal of glucose and insulin, where a high frequency describes a fast response of the regulatory system, meaning that the first glucose peak appears early, as seen in the blue and red lines in Figure 1. The amplitude (A) of the undamped sinusoidal depends on the body’s tolerance of the ingested glucose and relates to the height of the glucose peak together with the damping coefficient (α). The yellow dynamic has a larger amplitude than the blue and red ones, and the larger damping coefficient in the red dynamics avoids the under- and overshoots seen in the blue dynamic Figure 1. It should be noted that insulin is not part of the solution in Equation (1) since it was eliminated in the derivation and described in terms of glucose and the estimated parameters. This makes the model very attractive to use in a setting when insulin cannot be measured, and CGM could be used to measure glucose, such as when performing measurements at home. Since no parameter directly describes the maximum postprandial glucose concentration, an expression of this was derived in Equation (2).
(2)$$G_{max}=G_{b}+\frac{e^{-\kappa \cos^{-1}\left(\kappa /\sqrt{1+\kappa^{2}}\right)}}{\sqrt{1+\kappa^{2}}}\;A\;\;\;\;\;\;\;\mathrm{where}\;\;\;\;\;\kappa =\frac{\alpha }{\omega }$$

The model was originally shown to fit OGTT data well, despite mechanisms such as the role of adrenal cortical and medullary function in glucose economy were not accounted for [27]. In the present study, the model was used to describe the postprandial glucose concentration for an MMTT where the subjects were fasted prior to ingesting the meal.

### 2.6. Statistical Analyses

The parameters of the model were estimated within the nonlinear mixed effects model framework [28]. Here, we refer to our nonlinear regression model Equation (1) as G*,* which depends on the individual model parameters $\varphi_{i}=\left\{G_{b,\left(i\right)},A_{i},\omega_{i},\;\alpha_{i}\right\}$ and regresses to the glucose measurements $y_{i}$ with some measurement or process error $\varepsilon ∼N\left(0,\sigma^{2}\right)$ with an assumed constant variance across observations (Equations (3) and (4)). Here, i represents the subject index.
(3)$$y_{i}=G\left(\varphi_{i},\;t\right)+\varepsilon_{\;}$$
(4)$$\varphi_{i}=\beta_{\;}e^{\eta_{i}+A_{\;}x_{i}{}_{\;}}$$

The individual parameters $\varphi_{i}$ are described by the fixed effects (shared among all individuals) β, and the random effects $\eta_{i}∼\overset{n-1}{\underset{j=1}{\sum }}\mu_{j}I_{\left\{i\in M_{j}\right\}}+N\left(0,\Omega \right)$. Additionally, they are affected by covariates $x_{i}$ via the covariate matrix A. Here, $\eta_{i}$ is a vector with four elements, representing the random effects (modeling the variation within the test population) on each of the four parameters in $\varphi_{i}$. The mean vector ($\mu_{j}$) of the multivariate normal distribution is dictated by what group $M_{j}$ the individual i is most likely to belong to via the indicator function $I_{\left\{i\in M_{j}\right\}}$, effectively making $\eta_{i}$ a mixture of n multivariate normal distributions and allowing the identification of subgroups (clusters) within the data. The covariance matrix is denoted by Ω**,** which we assume to be diagonal, i.e., no correlation between parameters. Associations between clusters and clinical parameters were investigated using one-way analysis of variance (ANOVA) and Chi-squared tests for continuous and categorical data, respectively.

The association between response clusters and gut microbiota was investigated using ANOVA on selected genera (explained in the fecal microbiota section) that were reported to associate with glycemic regulation. Additionally, we measured the Pearson correlation between estimated model parameters that separate response clusters and t-tests to assess the significance of the correlation. The intervention effect on gut microbiota composition was investigated using log fold change in the baseline and 12 wk. on all species using random forest analysis within a repeated double cross-validation framework [29].

The parameter estimation software Monolix was used to simultaneously estimate the random and fixed effects (Monolix 2021R2, Lixoft SAS, a Simulations Plus company, Lancaster, CA, USA). Covariates were imposed to reduce the variance not reflecting blood glucose control. Age and site were imposed as covariates on the baseline ($G_{b}$). Since the breakfast meals given to the two treatment groups had similar nutritional composition (Table 1), treatment (high or low GI) was imposed as a covariate on the amplitude (A), thus accounting for differences in MMTTs between treatments. Parameters were estimated per individual and occasion (pre- and post-trial). The relative standard error (RSE) was used as an estimate of the uncertainty in the estimated parameters as follows:(5)$$RSE=100\cdot \frac{e_{std}}{\hat{y}}.$$

Here, $e_{std}$ describe the standard error and $\hat{y}$ is the estimate. We consider an RSE value below 50% percent a valid estimate using the Monolix software.

## 3. Results

In total, 155 individuals completed the two MMTTs and OGTTs (baseline and wk. 12) and were included in the analyses. Calculations on fecal microbiota were based on 130 individuals who provided two fecal samples within the participants that performed the two MMTT and OGTTs (baseline and wk. 12) (Table 2).

### Postprandial MMTT Glucose Responses

Individual parameters of the kinetic model (baseline, amplitude, damping, and frequency) from Equation (1) were estimated using the postprandial MMTT glucose response at baseline and wk. 12. The parameters were successfully estimated with RSE < 43% in all cases, which indicated certainty in the estimates. Variation among individuals resulted only in the parameters amplitude $A_{i}$, frequency $\omega_{i}$, and baseline $G_{b,\left(i\right)}$ since random effects on the damping parameter were not estimated with enough certainty. Clusters were therefore not based on the damping parameter $\alpha_{i}$. The covariate group membership (high-GI or low-GI) could also not be estimated with enough precision, meaning that there was no effective difference in the response of the two meals. The model fitted well to the response of the breakfast MMTT, as given by the low RSE, although some systematic phenomena could not be captured, e.g., the slow undershoot in subject 104 (Figure 2).

Two plasma glucose concentration profile clusters (A and B) were successfully identified (RSE < 33%), which were well separated in the amplitude and frequency parameters but not in the baseline parameter, although the cluster membership was estimated in the baseline parameter as well (Figure 3). Estimating more than two clusters rendered the cluster membership parameter unidentifiable, and a clear separation of individuals was visible when using a single lognormal distribution instead of a mixture of lognormal distributions. This led to the choice of estimating the likelihood of cluster membership using two lognormal distribution modes (clusters). The distribution of the clusters in the covariates is shown in Appendix A.

The individuals in cluster A had, in general, a higher frequency $\omega_{i}$ and a lower amplitude $A_{i}$ (Figure 3). This is also visible in the postprandial glucose profiles of the MMTT data when participants were color-coded by cluster membership (Figure 4). Individuals in cluster A had, in general, a lower peak in plasma glucose response. The peak also appeared later for cluster B (which equates to a lower frequency $\omega_{i}$ of the sinusoidal function in Equation (1)). The clusters consisted of approximately 46% of cluster A and 54% of cluster B.

The clusters at baseline were associated with known diabetic risk markers such as HbA1c (*p* = $2.8\times 10^{-5}$), insulin sensitivity indices (QUICKI (*p* = $1.4\times 10^{-6}$), Stumvoll (*p* = $1.7\times 10^{-3}$), Matsuda (*p* =$\;1.8\times 10^{-8}$), and waist circumference (*p* = $1.1\times 10^{-6}$) using one-way ANOVA (Figure 5). Although the clusters mostly involved amplitude and frequency, all the parameters correlated with the risk factors (Appendix A).

Importantly, the clusters also associated differently with conditions reflecting clinical cut-offs for differential glucose control, i.e., prediabetes (fasting HbA1c ≥ 5.7% and fasting blood glucose > 100 mg/dL, *p* = 0.01 [30], insulin resistance (*p* =$\;6.5\times 10^{-7}$), (Matsuda index ≤ 2.5), and glucose control (*p* = $6.6\times 10^{-5}$) (normal, impaired, or diabetic [30,31]) using a Chi-squared test.

Most of the subjects classified as normoglycemic also belonged to cluster A, and most of the subjects classified as “impaired” or “diabetic” belonged to cluster B (Figure 6). The frequencies of glycemic control classes in each cluster are indicated in Appendix A.

The same analysis as described using the breakfast MMTT response before the intervention (at baseline) was made using the breakfast MMTT response post-intervention, where similar cluster memberships were identified (Figure 7). However, the average Euclidean silhouette measure decreased from 0.58 to 0.36, indicating that the clusters were more distinct using baseline data. Moreover, it was observed that 60% of the subjects in cluster B improved in their glucose regulation by a decreased amplitude and increased frequency parameter value when comparing baseline values with those after 12 weeks of intervention. Additionally, in the low GI group, 58% decreased their baseline parameter, 61% decreased their amplitude parameter, and 59% increased their frequency parameter after the intervention compared to baseline, which all relate to improved glycemic control. In comparison, in the high GI group, 49% decreased their baseline parameter, 46% decreased their amplitude parameter, and 57% increased their frequency parameter compared to baseline. However, there was only a statistically significant change between the two time points in the low GI group in the amplitude parameter (Wilcoxon signed-rank test *p* = 0.001). Furthermore, within cluster A, there was a minor enrichment of women (63 and 64% pre- and post-intervention, respectively) compared to the entire study population (53% women). Hence, sex differences were associated with the cluster membership (*p* = 0.001, using Chi-squared test), QUICKI (*p* = $9\cdot 10^{-4}$, ANOVA), and Matsuda (*p* = $\;8.9\cdot 10^{-6}$). Furthermore, we found a weak association (*p* = 0.046, Chi-squared test) between improvement (lowering) of the baseline parameter $G_{b}$ and sex differences, where women improved their baseline parameter more than men independently of the treatment group.

Some of the individuals changed cluster from pre- to post-trial (~26% change in each cluster), but there was no significant difference between clusters (Cohen’s kappa = 0.42, moderate stability between clusters (95%CI 0.27–0.56)). The change in parameters from baseline to wk. 12 (change = baseline − wk. 12) did not correlate significantly with the change in risk markers.

Interestingly, the identified plasma glucose response clusters at baseline were associated with the gut microbiota genera *Clostridium sensu stricto 1* (ANOVA *p* = 0.007) and *Blautia* (ANOVA *p* = 0.024). However, these genera correlated weakly with the estimated amplitude (ρ = −0.2, *p* = 0.02 and ρ = 0.2, *p* = 0.02, respectively) and frequency (ρ = 0.14, *p* = 0.08, and ρ = −0.19, *p* = 0.02, respectively) parameters that separated the clusters. Here, ρ denotes the Pearson correlation coefficient, and *p* denotes the probability that the correlation is zero, using a t-test. As expected, we found no differences in gut microbiota composition between the two intervention groups (log fold change to predict low vs. high GI, balanced error rate 0.5). However, there was a clear difference attributed to the site (balanced error rate 0.09), and a difference in the Shannon diversity index (ANOVA *p* = 0.0134) between study centers was also noted (Appendix A).

## 4. Discussion

To dissect glucose data into features representing postprandial events, we used a model with only four parameters to identify clusters from standardized breakfast meal tolerance test responses that strongly related to T2DM risk factors. Although the model did not capture all systematic variation in the data, it was flexible enough to allow the identification of differential glycemic response clusters after mixed meal tests that were differentially associated with risk factors of T2D and gut microbiota. The results suggest that a standardized breakfast meal could provide meaningful data to predict risk factors of T2DM from dynamic glucose response measurements.

Plasma glucose response clusters were mostly separated in the amplitude and frequency parameters and not in the baseline glucose parameter (Figure 3 and Figure 7), which confirms that a dynamical model captures more information than a fasting plasma measurement and can more effectively be used for prediction of glycemic regulatory status. Individuals in cluster A were deemed to have better glucose control since they were characterized by a more favorable T2DM risk marker profile: a lower glycemic response, lower amplitude, and higher frequency parameters, whereas cluster B had the opposite traits and was therefore more likely to develop health issues related to their glycemic control such as T2DM. However, the individuals classified as “diabetic” (Figure 6) may be on the borderline to be classified as diabetics since this classification was using data from one OGTT when all participants were classified as non-diabetics at screening. Predictive tests of glycemic control classifications from estimated parameters were not analyzed due to the imbalance among classes, i.e., a lack of diabetic patients in the data. Furthermore, some patients in the healthier cluster A had impaired glycemic regulation, indicating that the estimated parameters give more information about the patient’s glycemic regulation than solely a classification based on OGTTs.

The breakfast MMTT clusters did not associate with the study site nor with treatment, which suggests that using these as covariates successfully captured their variance in the data. However, the low GI group improved more than the high GI group in their estimated parameters (decreased average baseline, decreased average amplitude, and increased average frequency) after a 12-week intervention, which suggests that the low GI diet aided in improving the glycemic control of the participants [13]. This could also explain why the cluster separation reduced from an average silhouette value of 0.58 to 0.36 from baseline to post-intervention, although there was no statistical difference overall between the two time points (Wilcoxon signed-rank test). Sex differences could not be estimated as a covariate to the dynamic model with enough precision, but response clusters were associated with gender, where cluster A consisted of more women, both at baseline and after 12 wks. This lack of change in gender distribution across the intervention and the fact that women improved their glucose baseline independently of treatment (high or low GI) indicates that the fasting glucose was not lowered in women due to treatment but merely due to their participation in the study. Interestingly, a study investigating the 8 h average plasma glucose concentration on data from the same MEDGI-Carb trial revealed that the high GI diet induced a higher glucose concentration (23%, *p* < 0.05) than the low GI diet in women [32]. However, they also concluded that the response to the breakfast meal alone did not show this difference; hence, their findings are in line with our results.

Our data suggest that a standardized breakfast MMTT based on regular foods may be an alternative to an OGTT, especially among patients with a high risk of nausea, such as pregnant women or bariatric surgery patients [33,34]. In contrast, an MMTT does not cause these side effects and is therefore an alternative to OGTT. Furthermore, our results based on a standardized breakfast meal including commonly consumed foods are in line with conclusions from a recent review that compared an OGTT to an MMTT and found a strong or very strong correlation (r = 0.9–0.97) between an OGTT and MMTT, which further supports that it may be used as an alternative to the OGTT [11]. Additionally, the metabolic feedback from an MMTT that includes all macronutrients (carbohydrates, fat, and proteins) provides more comprehensive information on glucose homeostasis compared to a single macronutrient [35,36]. However, the MMTT should be standardized and preferably provided as a breakfast to avoid complications of the glucose dynamics with lingering metabolic effects of other meals. If standardized, cluster membership could potentially be estimated at home using the MMTT and a CGM connected to a cellular device, as CGM data were shown to capture clusters of individuals based on glucose variability [7].

Previous studies have shown that gut microbiota is associated with postprandial glucose response [4,16,37], but no studies have investigated associations with response clusters. In our study, we found that glucose response clusters were associated with the bacterial genera *Clostridium sensu stricto 1* and *Blautia*. Cluster A had a higher proportion of *Clostridium sensu stricto 1* than cluster B and vice versa for *Blautia*, which is consistent with previously reported associations of these genera with glucose control [16,17,18]. Future studies should test how dietary interventions may affect these genera and reveal their mechanistic links with postprandial glucose response. As expected, there were no differences in microbiota composition between groups after intervention since the diets were similar except for low/high GI. In accordance with other studies, large differences between study centers were found, probably due to differences in dietary and lifestyle patterns as observed [13].

Our study has several strengths, including the large sample size with participants from three countries (Italy, USA, and Sweden), which reduces the chances that treatment effects or found clusters would be confounded by the cohort. In addition, the MMTT was robustly designed with participants carefully monitored during the test days and strictly standardized meal composition across the three centers to reduce the risk that the differences in response would be due to differences in intake. Furthermore, the mechanistic model gave interpretable clusters using only four identifiable parameters, which were estimated using a mixture of lognormal distributions. Although a mixture of distributions is rarely used, it proved useful in estimating the likelihood of cluster membership. However, although the method enabled investigating glucose control from the dynamical response, it should be noted that all descriptive variance was not captured using the model (e.g., slow undershoot).

Limitations included the fact that all participants were at risk of developing T2DM. Hence, although OGTT and MMTT responses were described using the same model to estimate insulin sensitivity [38], our method remains to be validated in a broader population, including patients with manifested T2DM and gestational diabetes. Also, the response to the lunch was not applied using this model since a more complex dynamic would be needed to account for the lingering response of the breakfast.

The MMTT used in the present trial was based on a Mediterranean diet. However, different diets and foods may have different effects on gastric emptying time and blood glucose response, which should be taken into consideration when designing future studies [39,40]. Validating our method in a cohort with a balanced set of individuals with normoglycemic, impaired glucose control, and diabetics may allow for the development of an algorithm to classify these states from the estimated parameters using our model on the response of MMTTs. Furthermore, this algorithm could potentially be used with a cellular device and a CGM in a home setting and be carried out periodically to facilitate tailored preventative treatment of prediabetes and T2DM.

## 5. Conclusions

We used a simple model to successfully describe glucose response to a standardized breakfast MMTT based on common foods and identified two response clusters that were associated differently with T2DM risk markers and gut microbiota. Future studies should investigate if such clusters can be identified by an algorithmic self-sampling tool for the classification of differential T2D risk profiles based on standardized breakfast MMTT in a home setting using continuous glucose monitoring and whether tailored diet and lifestyle advice may lower T2D risk.

## Figures and Tables

**Figure 1 nutrients-15-04369-f001:**
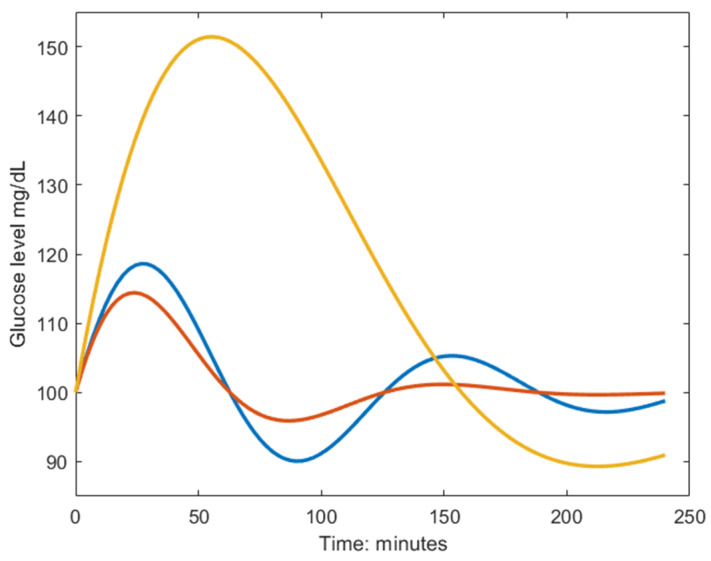
Example dynamics generated from the model in Equation (1). The blue curve is characterized by a fast biphasic response to the MMTT, thus having high frequency (ω) and low amplitude (A). The red curve has a larger damping coefficient (α) which yields a faster monophasic return to baseline. The yellow curve is characterized by a slow response to the MMTT, meaning poor glucose regulation, and is described by the inverse parameter relationship as the blue line but shares the same damping coefficient.

**Figure 2 nutrients-15-04369-f002:**
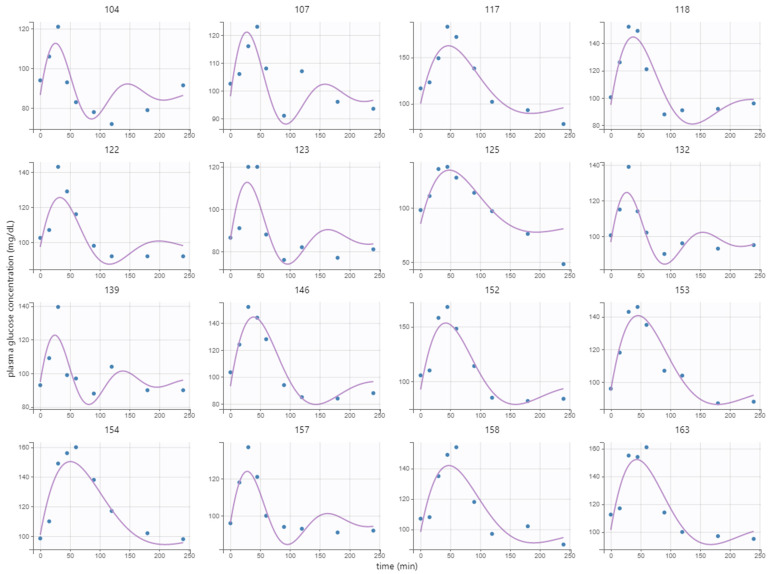
Model fit to postprandial breakfast MMTT response at baseline of 16 randomly selected representative subjects. Here, points represent measurements, and lines represent the fitted model values.

**Figure 3 nutrients-15-04369-f003:**
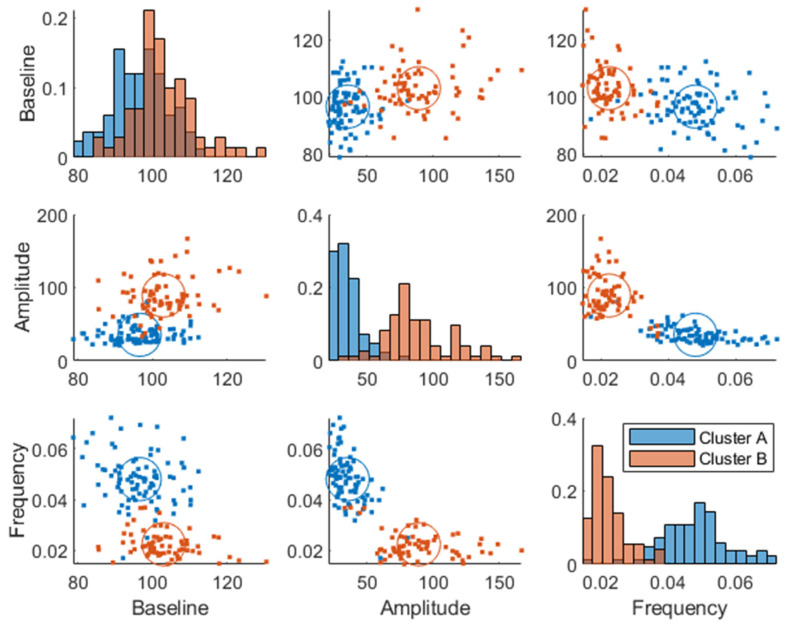
Joint parameter distribution obtained by fitting the model in Equation (1) to the postprandial breakfast MMTT data. The blue and red colors represent clusters A and B, respectively. The diagonal represents histograms of the parameter distribution (color-coded by transparent cluster color), and the off-diagonal represents pairwise joint distributions. Note that overlapping colors appear brown.

**Figure 4 nutrients-15-04369-f004:**
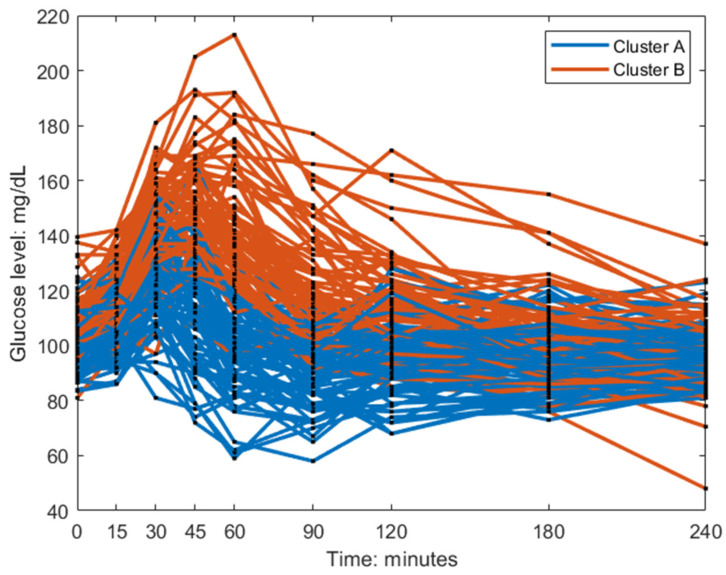
Baseline postprandial breakfast MMTT response color-coded by the clusters.

**Figure 5 nutrients-15-04369-f005:**
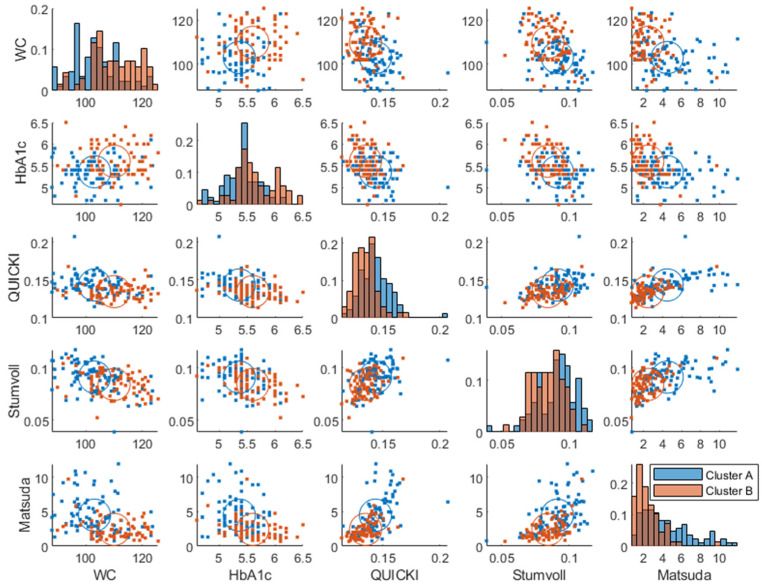
Baseline joint distribution of diabetes risk markers, which had significant associations with clusters. The diagonal represents histograms of the parameter distribution (color-coded by transparent cluster color), and the off-diagonal represents pairwise joint distributions. Note that overlapping colors appear brown.

**Figure 6 nutrients-15-04369-f006:**
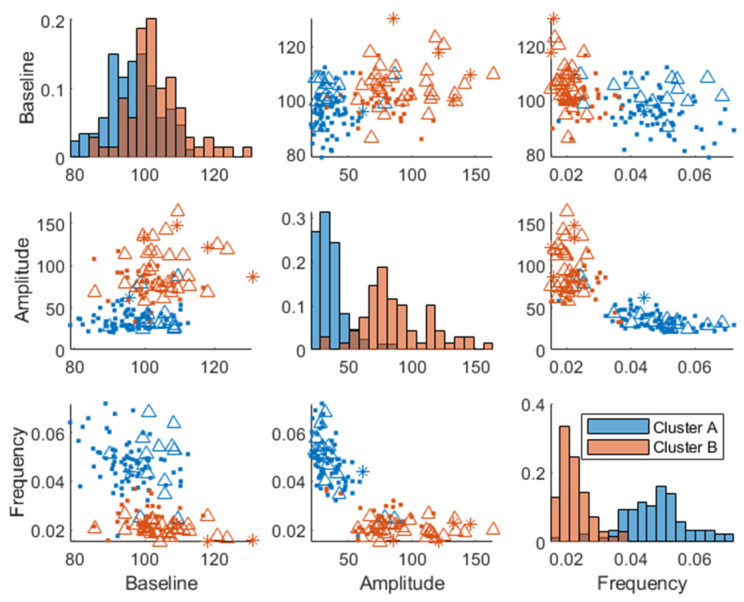
Baseline joint parameter distribution obtained by fitting the model in Equation (1) to the postprandial MMTT data. The different markers (dots, triangles, and asterisk) represent subjects classified as “normoglycemic”, “impaired” glucose control, or “diabetic”, respectively. Here, we used the OGTT measurement after 2 h and classified normal glycemic regulation as <7.7 mmol/L, impaired glucose tolerance in the range of 7.8–11.0 mmol/L, and diabetic ≥ 11.1 mmol/L. The diagonal represents histograms of the parameter distribution (color-coded by transparent cluster color), and the off-diagonal represents pairwise joint distributions. Note that overlapping colors appear brown.

**Figure 7 nutrients-15-04369-f007:**
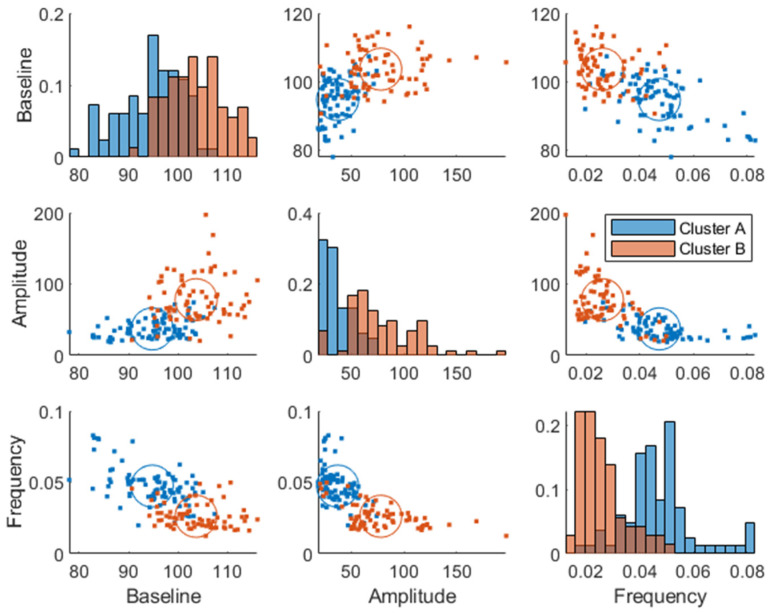
Week 12 joint parameter distribution obtained by fitting the model in Equation (1) to the postprandial breakfast MMTT data. The blue and red colors represent clusters A and B, respectively. The diagonal represents histograms of the parameter distribution (color-coded by transparent cluster color), and the off-diagonal represents pairwise joint distributions. Note that overlapping colors appear brown.

**Table 1 nutrients-15-04369-t001:** Food composition and nutrients of standardized meals.

**High GI Meal**							
**Foods Name**	**Serving Size (g)**	**Energy** **(Kilocalories)**	**Proteins (g)**	**Fat (g)**	**Total ** **Carbohydrates (g)**	**Soluble ** **Carbohydrates (g)**	**Total Dietary Fiber (g)**
Breakfast							
Cornflakes	30	140.4	2.5	0.3	26.4	4.0	1.5
Bread wholegrain, Pan Bauletto (Barilla)	24	64.0	2.0	0.9	11.4	1.7	1.1
Eggs, whole *	50	77.5	6.3	5.3	0.6	0.0	0.0
Extra virgin oil, olive	18	162.0	0.0	18.0	0.0	0.0	0.0
Ham, dry cured (country style), no visible fat eaten	85	52.2	7.5	2.0	0.5	0.0	0.0
Apple, fresh, without skin (Golden Delicious) *	150 *	78.0	0.4	0.3	20.7	20.7	3.6
Milk, 1% fat or low-fat, lactose-free	244	102.5	8.2	2.4	12.2	12.2	0.0
TOTAL		676.6	27.0	29.1	71.8	38.6	6.2
**Low GI Meal**							
**Foods Name**	**Serving Size (g)**	**Energy ** **(Kilocalories)**	**Proteins (g)**	**Fat (g)**	**Total ** **Carbohydrates (g)**	**Soluble ** **Carbohydrates (g)**	**Total Dietary Fiber (g)**
Breakfast							
Piadella (Mulino Bianco—Barilla)	75	255.0	5.6	8.4	38.3	2.3	2.0
Extra virgin oil, olive	10	90.0	0.0	10.0	0.0	0.0	0.0
Eggs, whole *	50	77.5	6.3	5.3	0.6	0.0	0.0
Ham, dry cured (country style), no visible fat eaten	38	60.9	7.7	3.1	0.0	0.0	0.0
Apple, fresh, without skin (Golden Delicious) *	150 *	78.0	0.4	0.3	20.7	20.7	3.6
Milk, 1% fat or low-fat, lactose-free	244	102.5	8.2	2.4	12.2	12.2	0.0
TOTAL		663.9	28.2	29.4	71.7	35.2	5.6

* Edible amount.

**Table 2 nutrients-15-04369-t002:** Baseline characteristics of the subpopulations analyzed in MMTT, OGTT, and fecal microbiota. There was no significant difference between treatment groups.

	High GI (MMTT and OGTT)	Low GI (MMTT and OGTT)	High GI (Fecal Microbiota)	Low GI (Fecal Microbiota)
Number of participants	72 (50% women)	83 (54% women)	57 (51% women)	73 (53% women)
Age (years)	55.8 ± 9.9	56.0 ± 10.5	57.0 ± 9.7	55.8 ± 10.7
BMI (kg/m^2^)	30.8 ± 3.0	31.1 ± 3.2	30.4 ± 3.1	30.9 ± 3.2
Waist circumference (cm)	107.3 ± 9.2	105.1 ± 8.6	106.5 ± 9.2	105.1 ± 8.4
Glucose (mg/dL)	105.5 ± 10.2	103.4 ± 10.3	106.4 ± 10.5	102.9 ± 10.2
Total cholesterol (mg/dL)	187.8 ± 30.8	192.2 ± 33.0	189.7 ± 30.5	192.7 ± 32.7
Triglycerides (mg/dL)	114.8 ± 44.6	122.2 ± 68.8	113.9 ± 45.5	117.6 ± 60.0
HDL (mg/dL)	48.4 ± 11.6	47.7 ± 11.8	50.2 ± 11.8	47.9 ± 11.8
LDL (mg/dL)	116.1 ± 27.6	119.8 ± 26.6	116.8 ± 27.4	120.6 ± 27.3
Systolic blood pressure (mm Hg)	124.6 ± 12.4	128.5 ± 13.7	124.1 ± 12.5	128.1 ± 13.8
Diastolic blood pressure (mm Hg)	80.9 ± 8.9	81.9 ± 8.5	81.1 ± 9.0	82.1 ± 8.6

## Data Availability

The data presented in this study are available upon reasonable request from the corresponding author.

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
