# Peer review of "Differential Responders to a Mixed Meal Tolerance Test Associated with Type 2 Diabetes Risk Factors and Gut Microbiota—Data from the MEDGI-Carb Randomized Controlled Trial"

_nutrients, 2023, doi:10.3390/nu15204369_

Round 1

Reviewer 1 Report

The authors present an observational study that aims to fit a simple mathematical model to dynamic postprandial glucose data obtained from repeated mixed meal tolerance tests (MMTTs) conducted among participants at an elevated risk of type 2 diabetes (T2DM). The primary objectives of the study are to identify response clusters and investigate their association with T2DM risk factors and gut microbiota.

The findings of the study indicate that model-based analysis of MMTTs conducted on a cohort of 155 participants (comprising 81 females and 74 males) revealed the presence of two distinct plasma glucose response clusters, which were found to be associated with baseline gut microbiota.

Comments:

1.

The abbreviation 'GI' for glycemic index is suggested to be indicated in the text.

2.

It is suggested to simplify Table 1 for better readability and provide the statistical test p-values for both the High GI and Low GI groups.

However, if it follows a parallel randomized controlled trial (RCT) design, it may be optional.

Nevertheless, providing this information, such as p-values, may be required due to the differing sample sizes in the MMTT and OGTT groups, as well as the fecal microbiota group.

3.

I don't fully understand how gut microbiota is calculated in this study, as well as the correlation between gut microbiota and clusters of A and B.

4.

Is there a difference in the model fit to postprandial breakfast MMTT responses at baseline between the high-GI and low-GI groups?

5.

Are Figures 3-7 used to define two distinct plasma glucose response clusters in the mixed High GI and Low GI groups' sample size?

If so, is it appropriate to cluster two distinct plasma glucose responses within these mixed groups, or should the authors test whether these two distinct plasma glucose response clusters exhibit consistent trends in the High GI and Low GI groups?

6.

The clusters at baseline were associated with known diabetic risk markers such as HbA1c (p=2.810^-5), insulin sensitivity indices (QUICKI (p=1.410^-6), Stumvoll (p=1.710^-3), Matsuda (p=1.810^-8)), and waist circumference (p=1.110^-6) using a one-way analysis of variance (ANOVA) in Figure 5.

But Figure 5 does not present the scatter plot and frequency distribution of Stumvoll?

It should be explained.

6.

Some statistical methods are mentioned in the text, including the t-test, Pearson correlation coefficient, and one-way analysis of variance (ANOVA), but the Statistical Analyses section does not provide detailed explanations.

The authors should provide this information.

7.

Figure 6 presents a similar distribution and clustering of normoglycemic, impaired, or diabetic individuals for clusters A and B. This seems to imply that clusters A and B are unable to distinguish between the three phenotypes: normoglycemic, impaired, or diabetic.

8.

The lack of difference between the Joint parameter distribution at Baseline in Figure 3 and the Joint parameter distribution at Week 12 in Figure 7 suggests what?

Author Response

Thank you very much for your thorough review. Attached you will find our answers to your comments.

Reviewer 2 Report

In this well-designed paper, the authors used a mathematical model to describe glucose response to a standardized breakfast (MMTT, mixed meal tolerance test), identifying two response clusters that were differently associated with T2DM risk markers and gut microbiota. All data comes from the MEDGI-Carb study, a multi-center randomized controlled trial, carried out in adults at elevated risk of developing T2DM.

The topic is very interesting, considering the increasing prevalence of T2DM worldwide; the identification of differential glycemic responders can help the development of better-tailored treatment for the prevention of prediabetes and T2DM. The manuscript is clear, well-structured, with the conclusions consistent with all the presented results.

 According to my opinion, a gender analysis is lacking, also considering the large sample size available. It will allows understanding the differences between men and women, suggesting more tailored treatments, and definitely improving the work.

Some other limitations, such as: the mathematical model that did not capture all systematic variation in the data, or the fact that all participants were at risk of developing T2DM, have been correctly reported by the authors in the text (page 13, lines 429-432; page 18, lines 508-511, respectively).

Some minor issues:

 Page 2, line 82. Please spell out the acronym “GI” the first time it appears

 Page 12, lines 400-401 “Furthermore, cluster A had a minor enrichment of women…..” I didn’t understand if the percentage of the women (63%) was higher or not respect to the entire study population (53%).       

Page 14, lines 481-482 “Previous studies…….is associated with postprandial glucose response,” please insert the references

             Table 1: please uniform “gram” and “g”  

Author Response

Thank you for your thorough review. Attached you will find the answers to your comments.

Round 2

Reviewer 1 Report

No further comment